# Data management and sharing: Practices and perceptions of psychology researchers

**John A. Borghi**[iD][1], Ana E. Van Gulick[2]*

**1** Lane Medical Library, Stanford University, Stanford, CA, United States of America, **2** University Libraries, Carnegie Mellon University, Pittsburgh, PA, United States of America

* anavangulick@cmu.edu

## Abstract

Research data is increasingly viewed as an important scholarly output. While a growing body of studies have investigated researcher practices and perceptions related to data sharing, information about data-related practices throughout the research process (including data collection and analysis) remains largely anecdotal. Building on our previous study of data practices in neuroimaging research, we conducted a survey of data management practices in the field of psychology. Our survey included questions about the type(s) of data collected, the tools used for data analysis, practices related to data organization, maintaining documentation, backup procedures, and long-term archiving of research materials. Our results demonstrate the complexity of managing and sharing data in psychology. Data is collected in multifarious forms from human participants, analyzed using a range of software tools, and archived in formats that may become obsolete. As individuals, our participants demonstrated relatively good data management practices, however they also indicated that there was little standardization within their research group. Participants generally indicated that they were willing to change their current practices in light of new technologies, opportunities, or requirements.

## Introduction

Interrelated issues including an overrepresentation of positive results [1, 2] unreported flexibility in analytical methods [3–5], and low levels of statistical power [6, 7] have resulted in uncertainty about the robustness of results described in the psychology literature. As researchers in and beyond the field have grappled with these issues, a variety of data stakeholders—including scholarly publishers, research funding agencies, and researchers themselves—have increasingly recognized data as an important research product.

In psychology, the lack of availability of the data and other materials underlying published results has been acknowledged for more than half a century [8–12] In at least one study, researchers reported that they could not make their data available because it was lost, inaccessible, or would require a considerable investment in time to make usable [13]. These results are consistent with complementary work examining barriers for data sharing in psychology [14] and as well as more broadly [15] and are illustrative of an important distinction—if datasets

**Data Availability Statement:** Our survey instrument, data, and notebooks containing the code used for analysis are openly available on the Dryad data repository: https://doi.org/10.5061/dryad.6wwpzgmw3.

**Funding:** The funders provided support in the form of salaries for authors [JAB: Stanford University; AEV: Carnegie Mellon University (through Jan 3, 2020), Figshare (beginning Jan 6, 2020)], but did not have any additional role in the study design, data collection and analysis, decision to publish, or preparation of the manuscript. The specific roles of these authors are articulated in the 'author contributions' section.

**Competing interests:** I have read the journal's policy and the authors of this manuscript have the following competing interests: Both authors work broadly in the field of data management and sharing. AEV is currently employed by the commercial company Figshare. This commercial affiliation does not alter our adherence to PLOS ONE policies on sharing data and materials. The project described in this paper was planned and all data was collected before AEV began her position at Figshare. The employers of neither author have influenced the development of the survey instrument, how data was collected or analyzed, or how the conclusions from this project are disseminated.

are to be regarded as an important product of the research process, then they must not only be made *available* (with access controls as appropriate), they must be made available in a *usable* form [16].

Befitting their position as a "leverage point in the research process" [17], scholarly publishers have begun to adopt a range of policies designed to ensure that authors provide appropriate access to the data underlying published work. In conjunction with the adoption of publication guidelines like TOP (Transparency and Openness Promotion) [18] and the use of open science badges to demarcate articles featuring pre-registered protocols and available materials, an increasing number of journals in and outside the field of psychology now require authors to complete a data availability statement describing either how the data underlying their article can be accessed or why the data cannot be made available. In 2015, the journal *Cognition* became the first major psychology journal to stipulate that the data underlying published work must be archived and made readily available at the time of publication [19]. Unfortunately, while journal data policies appear to increase the number of datasets that are reported to be available [20–22] data availability statements and other mechanisms do not guarantee that data will be made available in a truly publicly accessible or usable form. For example, evaluation of datasets made available under the *Cognition* data policy [23] and associated with articles given open science badges in *Psychological Science* [24] indicates that a non-trivial proportion were incomplete or not reusable in principle.

In addition to scholarly publishers, research funding agencies have also begun to institute policies related to research data. In response to a memo from the Office of Science and Technology Policy (OSTP), federal funding agencies in the United States have begun to require authors to submit data management plans (DMPs) as part of grant proposals. Though the requirements of such plans differ between and even within different agencies [25], generally such plans require researchers to identify the type(s) of data they plan to collect as well as how it will be organized, stored, and ultimately disseminated. Though there has been considerable emphasis on data management plans as a mechanism for ensuring the preservation of and access to scientific data, analyses of plans from the National Science Foundation (NSF) demonstrate that they frequently do not adequately describe the data researchers intend to collect or how it will be managed [26, 27] and, as with data availability statements, do not necessarily lead to data being shared in a complete or usable form [28].

Accompanying the implementation of publisher and funder-led data policies has been the development of infrastructure to facilitate researchers making their data available. Many academic institutions operate institutional repositories- archives designed to preserve and disseminate digital copies of, among other things, datasets and other research-related materials. Similarly, generalist open access repositories such as *figshare* (https://figshare.com/)

*Dryad* (https://datadryad.org/), and *Zenodo* (https://zenodo.org/) allow researchers to self deposit datasets so that they can be discovered, used, and cited by others. The *Open Science Framework (OSF)* (https://osf.io/) includes many features of a repository, but has additional functionality related to collaboration and project management- including preregistration of study plans. Complementing generalist repositories are those like *OpenNeuro* (https://openneuro.org/), which archive specific types of data (e.g. neuroimaging data). A recent study examining four decades worth of datasets associated with theses and dissertations in an institutional repository found that the majority were not accompanied by sufficient documentation to enable re-use [27]. To our knowledge there are no published studies examining the contents of general-purpose data repositories, however it is likely that the situation would be similar in any repository that does not feature a thorough curation process.

Ultimately, the responsibility for making sure research data is available in an usable form falls to researchers themselves. Recognizing the importance of research data requires

addressing a broad set of activities related to how it is stored, organized, and described throughout the entire research process [29]. Such activities are frequently grouped together under the umbrella term "Research Data Management" (RDM). As shown by visualizations such as the research data lifecycle [30, 31], RDM is an iterative and integral part of the research process—practices implemented early in the research process substantially affect what can be done later. While RDM-related activities are often described within the context of fostering open science practices [32], it is important to note that a dataset can be well managed even if it is not openly shared or the result of a study that was pre-registered, used open source tools, or published in an open access journal.

The American Psychological Association (APA) includes a statement about data availability in its Ethical Principles of Psychologists and Code of Conduct (section 8.14) [33] and the German Psychological Society (DGPs) has issued specific guidance about how psychology research data should be managed [34]. However, despite the development of policies and infrastructure encouraging researchers to follow data management best practices, information about the degree to which such practices are actually implemented by active researchers remains largely anecdotal.

Building on our previous study of data management practices among researchers using magnetic resonance imaging (MRI) [29] we sought to examine the data-related practices of researchers in the field of psychology. With the aim of being descriptive rather than prescriptive, our goal was to provide a detailed picture of data-related practices as they are actually applied in the day-to-day course of doing psychology research. In this way, we hoped to further efforts within and beyond the field to recognize data as an important research product.

## Methods

To investigate the data-related practices of psychology researchers, we adapted a survey developed as part of our previous study of neuroimaging researchers [29]. The survey was distributed via Qualtrics from January 25 to March 25, 2019. Before beginning the survey, participants were required to verify that they were at least 18 years old and gave their informed consent to participate. All study procedures were approved by the institutional review boards of both Carnegie Mellon University (STUDY2018_00000594) and Stanford University (E-Protocol 48696) and all research was performed in accordance with relevant guidelines and regulations.

### Survey design

The structure of our survey, which loosely follows the progression of a typical research project, drew from tools like the Data Curation Profiles [35] which have been used to characterize the data-related practices and needs of researchers working across a range of academic disciplines. As in our previous study, we consulted with active psychology and meta-science researchers as well as data management experts to ensure that each question was tailored to the specific terminology, practices, and tools employed by our participants. Reviewer feedback to our previous study also informed the survey development process, prompting the addition of questions to directly address issues related to training in data-related practices and the willingness of study participants to change their current practices in light of new technologies, opportunities, or requirements.

As in our previous study, our survey included several questions related to "data management maturity". This term was adopted from work based on the capability maturity model framework [36], which describe activities based on their degree of definition and standardization. Work invoking maturity models in the context of data-related practices has generally

focused on assessing the development of data-related services within organizations [37, 38]. In contrast, we asked participants to provide a quantitative maturity rating of both their own practices and the practices of the field of psychology as a whole.

In total, our revised survey included 64 multiple choice questions as well as an optional free-response field that participants could use to provide additional comments. Because we believed that our participants would come to our survey with different perspectives and degrees of expertise on data-related topics, an introduction at the start explained our goals and defined relevant terminology. The following sections were also accompanied by a description of specific activities and practices covered in that section.

The final section of the survey covered perceptions related to emerging publication practices. While some of these practices are not strictly related to promoting data as an important research product (e.g. posting preprints), they represent overlapping efforts to address concerns about the robustness and transparency of the research process. We did not collect information about participant age, gender, or institutional affiliation to preserve their anonymity.

### Distribution and filtering criteria

We sent an invitation to participate in the survey by e-mail to the corresponding authors of papers published in 40 high impact psychology journals in 2017 and 2018 (See S1 Appendix Journal List). After deduplication, e-mails were sent to a total of 8474 addresses using Mail-Chimp. The survey was also distributed via Twitter, psychology-related listservs, and Facebook groups such as PsychMAP and the Psychological Methods Discussion Group.

In order to capture a broad view of data-related practices and perceptions, the only inclusion criteria were that participants be active researchers in the field of psychology working with human participants, be at least 18 years of age, and consent to participate in the study. Data from participants who did not meet these criteria or who did not complete at least the first section of the survey were excluded from any subsequent analyses.

### Data analysis

The survey instrument, dataset, and Jupyter notebooks containing the code used for data analysis are openly available on the Dryad data repository [39]. Data were analyzed using the SciPy [40], NumPy [41], and Pandas [42] packages. Data visualizations were created using Matplotlib [43] and Seaborn [44].

## Results

### Participant characteristics

A total of 274 psychology researchers from 31 countries participated in our survey met our inclusion criteria. Though a precise response cannot be calculated because survey links were posted on social media, we speculate that the low response rate is due to the survey being distributed primarily through unprompted e-mails. Despite our low response rate, our survey captured responses from a broad range of individuals engaged in psychology research. As shown in Table 1, participants were affiliated with a variety of psychology subdomains, with the most common being Social and Personality, Cognitive, and Developmental Psychology. Including experience as an undergraduate research assistant, the participants in our sample reported that they possessed an average of 13.85 (SD = 8.76) years of experience actively doing psychology research. The majority of our participants indicated that they were affiliated with a predominantly research focused university (55.11%) followed equally by research and teaching focused universities or colleges (25.91%).

**Table 1. Participant characteristics.** Participants represented a range of psychology subdomains and included a mix of trainees (graduate students, postdoctoral fellows) and individuals with faculty positions (assistant, associate, and full professors).

| Professional Title | Percent | Research Area | Percent |
|---|---|---|---|
| Professor (US) or equivalent | 9.89% | Social and Personality Psychology | 22.63% |
| Associate Professor (US) or equivalent | 10.99% | Cognitive Psychology | 19.34% |
| Assistant Professor (US) or equivalent | 25.27% | Developmental Psychology | 16.06% |
| Adjunct Professor | 1.83% | Cognitive Neuroscience | 11.68% |
| Postdoctoral Fellow | 21.98% | Clinical Psychology | 10.95% |
| Graduate Student | 20.88% | Industrial/Organizational Psychology | 6.57% |
| Research Associate/Scientist | 5.13% | Biological Psychology | 4.01% |
| Project Coordinator or Lab Manager | 1.10% | Educational Psychology | 1.82% |
| Research Assistant | 1.47% | Health Psychology | 1.46% |
| Other | 1.47% | Quantitative Psychology | 0.73% |
| | | Other | 4.74% |

Participants indicated that they had received training related to research methods and data management from multiple sources, including classes during their undergraduate and graduate education (methods: 87.18%, data management: 33.33%), from other researchers who are in or collaborate with their research group (methods: 83.52%, data management: 64.10%), through self-education (methods: 74.36%, data management: 52.01%), and through guidance or best practice guidelines created by organizations or experts within the field of psychology (methods: 69.96%, data management: 34.80%).

For activities related to both data management (64.84%) and data sharing (71.06%), the majority of participants indicated that their institution either did not provide related support services or that they were unsure if such services were offered. By comparison, 45.79% of participants gave the same response for services related to information technology.

### Data management limits and motivations

As shown in Fig 1, when asked about what factors motivate their current data-related practices, participants gave relatively high ratings (using a 1–5 scale from 'not limited/motivated' to 'highly limited/motivated') to practical concerns such as the desire to prevent the loss of data (median = 5) and ensure continuity as their research team changes (median = 5) as well as broader concerns such as a desire to foster research transparency (median = 5) and reproducibility (median = 5). Ratings for factors that limit current practices were more diffuse, with lack of time (median = 3), professional incentives (median = 3), institutional support (median = 3), and training (median = 3) receiving relatively high ratings.

For questions related to data management maturity, participants were asked to rate the maturity of both their own practices and the practices of the field of psychology as a whole on a scale of 1 (ad hoc) to 5 (mature). Building from our previous study [29], we compared maturity ratings across the different phases of a research project as well as how participants rated the maturity of their own practices compared to their perceptions of the field as a whole. A series of Kruskal-Wallis H-tests [45] demonstrate that participants were significantly more likely to give higher maturity ratings to both their own ($H = 39.24$, $p < 0.000001$) and the field's ($H = 22.21$, $p < 0.000001$) practices related to data collection and analysis phases than those related to sharing. Similarly, the results of a series of Mann Whitney U tests [46] demonstrate that participants were significantly more likely to give higher ratings to their own practices related to data collection ($u = 27770.5$, $p < 0.000001$), data analysis ($u = 29030.0$, $p < 0.000001$), and data sharing ($u = 33280.0$, $p < 0.01$) than those of the field as a whole.

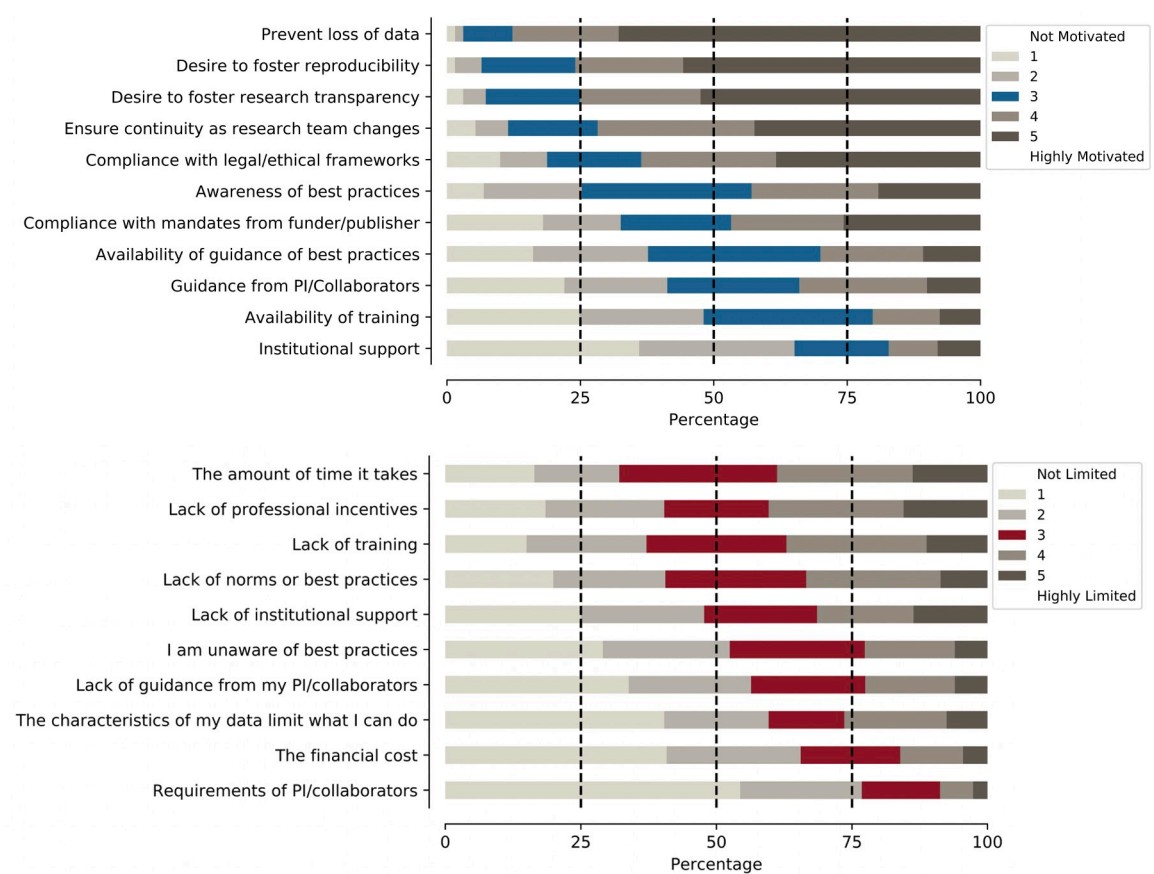

**Fig 1. Factors that motivate and limit psychology researchers' data management practices.** Participants were asked to rate the degree to which different factors motivate and limit their data-related practices on a scale of 1 (not limited/motivated) to 5 (highly limited/motivated). For motivations, participants gave high ratings to immediate and practical concerns, such as the desire not to lose data as well as broader concerns such as the desire to foster reproducibility and research transparency. Ratings for limitations were more diffuse.

## Data collection practices

As shown in Table 2, our participants indicated that they collect data through a wide variety of means and in a wide range of forms. In addition, they responded that their data needs to be preserved alongside a variety of additional material such as consent forms, text of question-naires, and information about the data collection session.

As shown in Table 3, participants reported using a variety of proprietary and open source software tools to build experiments, ask questions, and collect data from participants. A sub-stantial number of participants (42.31%) reported that they had deposited data collection-related code or syntax in the Open Science Framework in order to share it with others. Other options used for sharing included software specific hosting sites (e.g. GitHub) (20.81%), jour-nal articles (19.23%), websites (16.67%), and institutional repositories (10.26%).

Participants reported using a variety of methods to both organize and backup collected data. The most common methods for keeping data organized were the application of standard-ized file organization (53.16%) and naming (54.85%) schemes. However, 37.97% of partici-pants also responded that they follow general procedures for organizing their data that are not standardized or recorded and, when asked if everyone in their lab or research group uses

**Table 2. Data and related material.** Though the majority (86.13%) indicated that they collect primarily quantitative data, the participants in our sample indicated that they collect data in a wide variety of forms and formats that must also be accompanied by a diverse array of additional materials. "Other" data types included motion capture data, data from wearable devices, and data from administrative, institutional, and government records. "Other" forms of additional material included case summaries, payment information, and information about how questionnaires were modified.

| Data Types | Percent | Additional Material | Percent |
|---|---|---|---|
| Demographic Data | 97.06% | Informed consent-related documentation | 87.61% |
| Quantitative Data from Questionnaires | 87.39% | Text of questionnaires, scales, etc | 82.05% |
| Behavioral Data | 79.83% | Information about the data collection session | 76.07% |
| Qualitative Data from Questionnaires | 42.02% | Coding materials | 70.94% |
| Audio/Visual Recordings | 41.18% | Research-related stimuli | 69.23% |
| Clinical or Medical Data | 33.19% | Research protocol/paradigm related information | 64.53% |
| Neuroimaging Data | 30.25% | Computer code used for data collection | 64.53% |
| Physiological Data | 24.79% | Other | 1.71% |
| Eye tracking/Pupillometry Data | 24.79% | | |
| Neuropsychological or Aptitude Tests | 21.43% | | |
| Qualitative Data | 18.91% | | |
| Data from written documents | 10.92 | | |
| Genetic/Molecular Data | 10.50% | | |
| Other | 4.62% | | |

similar organizational systems, the majority (56.96%) of participants responded with either "No" or "I'm not sure".

Common methods for backing up or otherwise securing data included uploading data to the cloud (58.40%) and storing non-digital data in a secure location (55.88%). The majority of participants (55.04%) indicated that they keep two or more backup copies of their (digital) data though, when asked if everyone in their lab or research group uses the same systems for backing up their data, 48.42% responded with either "No" or "I'm not sure".

## Data analysis practices

As shown in Table 3 participants indicated that they use a wide variety of software tools to analyze their data, with the most common being Excel, SPSS, and R. The majority of participants also indicated that they write custom code themselves or adapt code written by others in order

**Table 3. Software tools used for data collection and analysis.** Software tools used to collect and analyze data. Parentheses for analysis software indicate the tools participants were taught to use as part of their education in research methods and statistics. "Other" responses for data collection software were largely comprised of survey tools (e.g. Survey Monkey, LimeSurvey) and tools for building and running behavioral experiments (e.g. Gorilla, JsPsych). "Other" responses for data analysis software largely consisted of neuroimaging-related tools (e.g. SPM, AFNI).

| Software | Percent | Analysis Software | Percent |
|---|---|---|---|
| Qualtrics | 57.45% | R | 67.38% (49.36%) |
| Matlab | 26.38% | SPSS | 63.52% (87.12%) |
| E-Prime | 22.13% | Excel | 48.50% (46.78%) |
| Custom Code | 22.13% | Matlab | 24.03% (21.46%) |
| Other | 22.13% | MPlus | 22.75% (24.03%) |
| PsychoPy | 19.57% | JASP | 13.30% (8.58%) |
| RedCap | 14.47% | Python | 10.73% (7.73%) |
| Inquisit | 9.36% | Other | 6.87%5 (6.87%) |
| None | 4.26% | Stata | 6.44% (8.15%) |
| | | Jamovi | 5.15% (0.86%) |

to analyze data. A substantial number of participants (41.20% of responding participants) indicated that they share such code using the Open Science Framework. Other popular methods for sharing included software specific repositories such as GitHub (19.74%), including custom code as part of a journal article (16.31%), and hosting code on a lab or project website (10.73%).

The majority of participants (53.88%) indicated that they document their activities during the data analysis phase of a project using a word processing or note-taking program while 23.38% indicated that they document their activities using a physical notebook or paper. The majority (63.47%) also indicated that everyone in their lab does not use similar systems for documentation. However, despite this lack of standardization, 58.80% of responding participants indicated that they believe that someone with a similar level of expertise could recreate their data analysis steps (including data cleaning and coding) from their documentation and notes without them being present.

## Data sharing practices

The majority of participants (62.88%) responded that they have archived, deposited, or published a dataset in order to make it available to others. The most common methods for doing so were via the Open Science Framework (44.54%) and as part of a journal article (33.19%). As shown in Table 4, the most common motivations for sharing data generally related to fostering research reproducibility, transparency, and validity. Participants also indicated that there was a wide variety of reasons why part or all of their data could not be shared, with the most common being the inclusion of sensitive data, the fact that their data may contain additional findings they wish to discover or describe in a publication, and the lack of institutional review board approval to share data.

In terms of data ostensibly made available upon request, 39.57% of participants stated that they have requested data associated with a paper or other scholarly publication from another researcher while 47.62% indicated that they have received such requests for their own data. Participants gave significantly higher ratings to their ability to send requested data associated with a publication in a usable form without significant effort than they did to their ability to actually use data they have received (u = 28544.5, p < 0.000001). When participants requested data from other researchers or sought openly accessible data, they most commonly reported that they used it to extend conclusions drawn from it or test alternative hypotheses (29.19%) or as part of completing a meta-analysis (25.41%).

**Table 4. Reasons for sharing and not sharing data.** Despite the growth of data sharing mandates from scholarly publishers, funding bodies, and other stakeholders, participants more frequently responded that their sharing was motivated by broader reasons, such as a desire to foster reproducibility, transparency, validation, and re-use. "Other" motivations for sharing data included giving other investigators a chance to make discoveries, establishing an open science track record, and as a way to backup data. "Other" reasons for not sharing data included difficulty understanding how data sharing works in the context of tribal agreements and the cumbersome nature of sharing large files.

| Reasons for sharing | Percent | Reasons for not sharing | Percent |
|---|---|---|---|
| To foster transparency and reproducibility | 67.40% | My data contains confidential or sensitive information | 50.68% |
| To allow other researchers to evaluate the validity of my conclusions | 59.47% | My data contains additional findings I wish to discover. | 40.36% |
| To communicate my results and/or add to the scholarly literature | 56.39% | Did not receive institutional review board approval. | 31.40% |
| To foster re-use | 55.50% | My supervisor/collaborators do not wish to share the data | 22.87% |
| It is mandated by a funder, publisher, or my institution | 27.31% | It would take too much time or effort to share. | 22.42% |
| Professional incentives | 18.50% | I do not know how to share my data | 8.07% |
| To establish intellectual property of patent claims | 1.32% | My data is proprietary or subject to IP concerns | 6.28% |
| Other | 3.96% | Other | 2.24% |

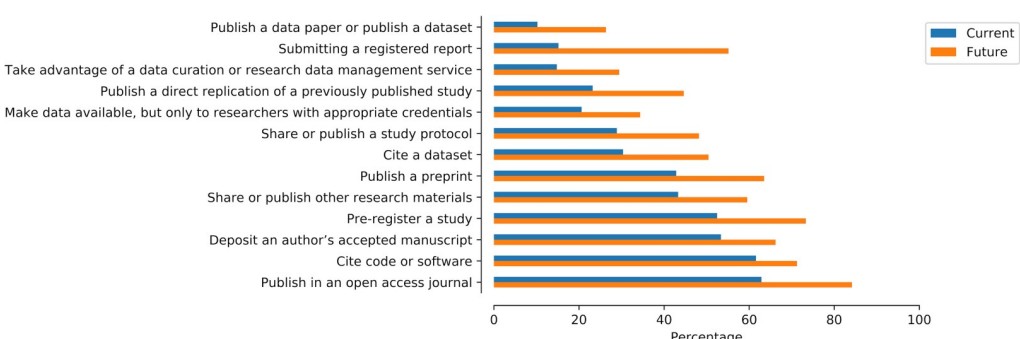

**Fig 2. Current and future adoption of emerging publication practices.** Percentage of participants who have (blue) and plan to (orange) adopt a range of scholarly communications practices.

In terms of the long-term archiving of data, the majority of participants (67.98%) stated that they (or their lab) typically keeps data for eight or more years, though 63.23% of those indicated that they do so in formats that may become obsolete in the future.

## Emerging publication practices

The majority of participants (59.91%) indicated that they are not limited in addressing their research questions by a lack of access to research data collected by others. Similarly, a substantial amount (53.30%) also indicated that they either do not or are not sure if they consider data to be a "first class" research product. However, as shown in Fig 2, our results indicate that the field of psychology is at a point of transition. While relatively few participants indicated that they are currently engaging in emerging scholarly communications-related activities, a larger number indicated that they plan to engage in such activities in the future.

## Discussion

In order to inform ongoing efforts within and beyond the field of psychology to recognize data as an important research product, we surveyed active psychology researchers about their data-related practices. Previous work surveying data-related practices in psychology has generally focused on practices and perceptions related to open science [47] and data sharing [14, 48]. However, for data and other materials to be efficiently examined, evaluated, and built upon, they must first be properly documented, organized, and saved. For this reason, we surveyed researchers about their data-related practices throughout the research process—during the data collection, analysis, and sharing phases of a project.

Our results demonstrate the complexity of managing and sharing data in a field like psychology. Data is collected from human participants using a diverse array of methods and tools, datasets often need to be preserved alongside a host of other research materials (e.g. surveys, stimulus sets, code used for data collection and analysis), and there is little consensus about how study procedures and analysis steps should be recorded and communicated. Complicating matters further, relatively few of our participants reported that they had received any formal training in data management or had made use of data management-related support services at their institution.

As individuals, our participants generally indicated that they apply reasonably good data management practices throughout the research process, with most responding that they have some system in place for backing up data, keeping files organized, and documenting what steps they go through during the analysis process. In line with the results of our previous study

of the neuroimaging community, participants in this study rated their own practices as significantly more mature than those of the field of psychology as a whole. It is possible that this trend simply reflects a well-known phenomenon of participants rating themselves as above average across a range of characteristics [49], but it is also possible that a study, which was explicitly advertised as being about data management practices in psychology, disproportionately attracted psychology researchers who actually do practice better than average data management to participate.

Unfortunately, also in line with our previous study, our participants frequently responded that their practices were not the same as others in their lab or research group. While the needs of different projects may necessitate the application of slightly different practices and procedures, this trend indicates that there is substantial room for improvement in terms of how well data management practices are applied at the lab or even project level. Instilling good data management involves more than simply having a set of practices related to organizing, saving, and documenting the data related to a particular project. It also involves ensuring that those practices are well described, clearly communicated, and followed by members of the research team.

This study was focused on the practices of researchers in the field of psychology, but our results should not be interpreted as a criticism or singling-out of the field. Though there has been improvement, the lack of availability of data underlying published work has been demonstrated to be an ongoing issue across the social [50] and biomedical [51] sciences. Similarly, surveys of data management practices among researchers at individual institutions have demonstrated that, while there may be substantial differences in the practices of researchers working in different areas, there is almost always room for improvement [52]. In general, communities with mature practices related to data management, such as high energy physics [53] appear to be the exception rather than the rule.

Psychology is actually well positioned to improve its data management practices and more fully recognize data as an important research product. As our results show, psychology researchers have already begun to adopt a variety of emerging scholarly communication, open science, and reproducibility-related practices such as publishing datasets and pre-registering study plans, that, in order to be effective, require good data management. More generally, the participants in our study indicated that they would be willing to change their data management practices throughout the data collection, analysis, and sharing phases of their research. Our survey results help show the current state of data-related practices in psychology and they also demonstrate that what is missing are incentives to change practices and knowledge of how to do so effectively.

Ultimately, while there are many stakeholders involved in setting related policies and requirements [54], responsibility for ensuring data is well managed falls to researchers during the day-to-day course of doing research. Incorporating data management into research methods coursework may represent one avenue for encouraging best practices. Though there is evidence from other fields that training in data management remains an "unmet need" [55, 56], discussion of data-related best practices would fit neatly into coursework that already covers research ethics, experimental design, and results reporting [57].

A full accounting of data management best practices is beyond the scope of this paper, but we have outlined several suggestions for researchers to consider in a supplementary document [58]. We would also strongly encourage researchers, in and outside of psychology, to consider the most immediate advantages of good data management. Even for researchers who are unwilling or unable to make their data openly available, implementing good data management practices will help make collaboration more efficient and prevent data—and investment in time and effort from both researchers and study participants—from being lost. For researchers

who can share, making data available through a trusted repository may have immediate and tangible benefits, such as increased citations [59] but the implementation of good data management practices is necessary for data to be shared efficiently and effectively. Data is an important product of the research process and fully recognizing its importance means addressing behaviors and practices that occur during the day-to-day course of doing research during the entirety of the research process.

## Supporting information

**S1 Appendix. Journal list.** Our survey was sent to the corresponding authors of papers published the following 40 high impact psychology journals between 2017 and 2018.
(CSV)

## Acknowledgments

The authors would like to thank the psychology researchers, data librarians, and meta-researchers who gave feedback on the survey instruments.

## Author Contributions

**Conceptualization:** John A. Borghi, Ana E. Van Gulick.

**Data curation:** John A. Borghi.

**Formal analysis:** John A. Borghi.

**Methodology:** John A. Borghi, Ana E. Van Gulick.

**Writing – original draft:** John A. Borghi, Ana E. Van Gulick.

**Writing – review & editing:** John A. Borghi, Ana E. Van Gulick.

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
