## [Decision Letter · Decision Letter 0]

24 Feb 2021

PONE-D-20-34753

­­­­­­­­Data Management and Sharing: Practices and Perceptions of Psychology Researchers

PLOS ONE

Dear Dr. Van Gulick,

Thank you for submitting your manuscript to PLOS ONE. After careful consideration, we feel that it has merit but does not fully meet PLOS ONE’s publication criteria as it currently stands. Therefore, we invite you to submit a revised version of the manuscript that addresses the points raised during the review process.

Reviewers have indicated that this paper is well-presented and addresses an important and current problem.  Mostly, they indicated that the analysis was acceptable, but there were a few requests for clarification and a few minor errors that need to be fixed.

We look forward to receiving your revised manuscript.

Kind regards,

Hussein Suleman, PhD

Academic Editor

PLOS ONE

Journal Requirements:

"I have read the journal's policy and the authors of this manuscript have the following competing interests: Both authors work broadly in the field of data management and sharing. This project described in this paper was planned and all data was collected before AVG began her position at Figshare. The employers of neither author have influenced the development of the survey instrument, how data was collected or analyzed, or how the conclusions from this project are disseminated."

We note that one or more of the authors are employed by a commercial company: Figshare.

3.2. Please provide an amended Funding Statement declaring this commercial affiliation, as well as a statement regarding the Role of Funders in your study. If the funding organization did not play a role in the study design, data collection and analysis, decision to publish, or preparation of the manuscript and only provided financial support in the form of authors' salaries and/or research materials, please review your statements relating to the author contributions, and ensure you have specifically and accurately indicated the role(s) that these authors had in your study. You can update author roles in the Author Contributions section of the online submission form.

3.2. Please also provide an updated Competing Interests Statement declaring this commercial affiliation along with any other relevant declarations relating to employment, consultancy, patents, products in development, or marketed products, etc.  

4. We noted in your submission details that a portion of your manuscript may have been presented or published elsewhere.

"The submitted version of this paper has been uploaded to PsyArXiv.

Borghi, J.A. & Van Gulick, A.E. (2020). Data Management and Sharing: Practices and Perceptions of Psychology Researchers. PsyArXiv, " ext-link-type="uri" xlink:type="simple">https://doi.org/10.31234/osf.io/7g3ae"

Please clarify whether this publication was peer-reviewed and formally published. If this work was previously peer-reviewed and published, in the cover letter please provide the reason that this work does not constitute dual publication and should be included in the current manuscript.

Reviewers' comments:

Reviewer's Responses to Questions

**Comments to the Author**

1. Is the manuscript technically sound, and do the data support the conclusions?

Reviewer #1: Yes

Reviewer #2: Yes

2. Has the statistical analysis been performed appropriately and rigorously? 

Reviewer #1: Yes

Reviewer #2: Yes

3. Have the authors made all data underlying the findings in their manuscript fully available?

Reviewer #1: Yes

Reviewer #2: Yes

4. Is the manuscript presented in an intelligible fashion and written in standard English?

Reviewer #1: Yes

Reviewer #2: Yes

5. Review Comments to the Author

Reviewer #1: This study describes a survey of psychology researchers’ data management and sharing. The methodology is well-described and appears to be sound, the results are clearly described and interpreted, and the potential for participation bias (which seems to be at play here) is rightly noted as a limitation. I believe this represents a solid contribution to the literature on research data management and sharing with a spotlight on psychology, which, as the authors note, is in a time of transition regarding research and data practices, partly motivated by reproducibility concerns.

I have two questions:

1. I was impressed that 8,474 unique researchers had been invited to participate in the survey, but I was disappointed to see that only 274 completed the survey, which represents merely a ~3% response rate. Could the authors speculate on why their response rate was so low? Does MailChimp provide any data on how many invitation emails were actually received and/or opened? Did many respondents start but not complete the survey?

2. The areas where inferential statistics were performed come across as being somewhat “cherry-picked”. Could the authors provide better motivation for these statistical analyses?

Minor points:

Line 174, “After filtering”: I’m not sure what this means. Does this mean “After deduplication”?

Figure 1: I recommend providing verbal labels in the graph legends rather than numerical labels (e.g., 1, 2, 3, . . . ) to faciliate reader comprehension.

Reviewer #2: This is an excellent paper that is well structured and that shows in a transparent way the study carried out, the data collected and the results and conclusions drawn. It complements existing studies on data management / sharing practices well, yet adds new insights on tools and practices used, and the maturity rating of data practices.

Minor correction needed in line 296: Figure 4 should be Table 4.

As suggestion: it would be good to visualize the information in tables 2, 3, 4 as graphs.

6. PLOS authors have the option to publish the peer review history of their article (what does this mean?). If published, this will include your full peer review and any attached files.

Reviewer #1: No

Reviewer #2: **Yes: **Veerle Van den Eynden

---

## [Author Response · Author response to Decision Letter 0]

16 Apr 2021

We have confirmed that our manuscript conforms to the style guidelines and that the reference list is complete and correct. 

The Competing Interests Statement and Funding Statement have both been updated for clarity as requested in our cover letter. 

Competing Interests Statement: 

I have read the journal's policy and the authors of this manuscript have the following competing interests: Both authors work broadly in the field of data management and sharing. AEV is currently employed by the commercial company Figshare. This commercial affiliation does not alter our adherence to PLOS ONE policies on sharing data and materials. The project described in this paper was planned and all data was collected before AEV began her position at Figshare. The employers of neither author have influenced the development of the survey instrument, how data was collected or analyzed, or how the conclusions from this project are disseminated.

Funding Statement: 

The funders provided support in the form of salaries for authors [JB: Stanford University; AEV: Carnegie Mellon University (through Jan 3, 2020), Figshare (beginning Jan 6, 2020)], but did not have any additional role in the study design, data collection and analysis, decision to publish, or preparation of the manuscript. The specific roles of these authors are articulated in the ‘author contributions’ section.

We have clarified that our manuscript was only posted on a preprint server and has not been peer-reviewed nor published in another journal. 

The initially submitted version of this paper was uploaded to the preprint server PsyArXiv:

Borghi, J.A. Van Gulick, A.E. (2020). Data Management and Sharing: Practices and Perceptions of Psychology Researchers. PsyArXiv, https://doi.org/10.31234/osf.io/7g3ae

Preprints on PsyArXiv are not peer-reviewed nor considered to be formally published, thus this submission does not constitute dual publication. 

Supporting information files 

We have added revised the name of our supplemental file on line 156 and added a caption starting on line 432.

Response to Reviewers

I was impressed that 8,474 unique researchers had been invited to participate in the survey, but I was disappointed to see that only 274 completed the survey, which represents merely a ~3% response rate. Could the authors speculate on why their response rate was so low? Does MailChimp provide any data on how many invitation emails were actually received and/or opened? Did many respondents start but not complete the survey?

Thank you for this question, we have adjusted the language in lines 175 to 181 clarifying that we believe our strategy of sending unprompted e-mails is responsible for the low response rate. We do not have data on how many e-mails were actually received, but we believe this number is still relatively high given that we used e-mails scraped from articles published 1 or 2 years prior to the distribution of our survey. In our discussion section (line 373) we note how our survey – which was explicitly advertised as being about data management in psychology – may have disproportionately attracted psychology researchers with better than average data management practices.

The areas where inferential statistics were performed come across as being somewhat “cherry-picked”. Could the authors provide better motivation for these statistical analyses?

We have added a sentence (line 227) clarifying that these analyses were in line with those conducted in our earlier survey.

Line 174, “After filtering”: I’m not sure what this means. Does this mean “After deduplication”?

This refers to the number of participants who met our inclusion criteria. We have revised this sentence to clarify this point.

Figure 1: I recommend providing verbal labels in the graph legends rather than numerical labels (e.g., 1, 2, 3, . . . ) to facilitate reader comprehension.

Thank you for this comment, we did not provide verbal labels for every value in survey text, so we did not feel it appropriate to add them here. However, we have adjusted the figure legends to further facilitate reader comprehension.

Minor correction needed in line 296: Figure 4 should be Table 4.

Thank you. We have corrected this error, now on line 306.

As suggestion: it would be good to visualize the information in tables 2, 3, 4 as graphs.

Thank you for this suggestion, we decided to use tables to communicate this information rather than graphs because we felt that a graph would not add any additional information.

---

## [Decision Letter · Decision Letter 1]

10 May 2021

­­­­­­­­Data Management and Sharing: Practices and Perceptions of Psychology Researchers

PONE-D-20-34753R1

Dear Dr. Van Gulick,

We’re pleased to inform you that your manuscript has been judged scientifically suitable for publication and will be formally accepted for publication once it meets all outstanding technical requirements.

Kind regards,

Hussein Suleman, PhD

Academic Editor

PLOS ONE

Additional Editor Comments (optional):

Reviewers' comments:

Reviewer's Responses to Questions

**Comments to the Author**

1. If the authors have adequately addressed your comments raised in a previous round of review and you feel that this manuscript is now acceptable for publication, you may indicate that here to bypass the “Comments to the Author” section, enter your conflict of interest statement in the “Confidential to Editor” section, and submit your "Accept" recommendation.

Reviewer #1: All comments have been addressed

Reviewer #2: All comments have been addressed

2. Is the manuscript technically sound, and do the data support the conclusions?

Reviewer #1: (No Response)

Reviewer #2: Yes

3. Has the statistical analysis been performed appropriately and rigorously? 

Reviewer #1: (No Response)

Reviewer #2: Yes

4. Have the authors made all data underlying the findings in their manuscript fully available?

Reviewer #1: (No Response)

Reviewer #2: Yes

5. Is the manuscript presented in an intelligible fashion and written in standard English?

Reviewer #1: (No Response)

Reviewer #2: Yes

6. Review Comments to the Author

Reviewer #1: (No Response)

Reviewer #2: (No Response)

7. PLOS authors have the option to publish the peer review history of their article (what does this mean?). If published, this will include your full peer review and any attached files.

Reviewer #1: No

Reviewer #2: **Yes: **V. Van den Eynden

---

## [Editor Report · Acceptance letter]

12 May 2021

PONE-D-20-34753R1 

­­­­­­­­Data Management and Sharing: Practices and Perceptions of Psychology Researchers 

Dear Dr. Van Gulick:

I'm pleased to inform you that your manuscript has been deemed suitable for publication in PLOS ONE. Congratulations! Your manuscript is now with our production department. 

Kind regards, 

on behalf of

Dr. Hussein Suleman 

Academic Editor

PLOS ONE